Characterisation of GLUT4 trafficking in HeLa cells: comparable kinetics and orthologous trafficking mechanisms to 3T3-L1 adipocytes

Morris Silke 1
Geoghegan Niall D. 2
Sadler Jessica B.A. 1
http://orcid.org/0000-0001-7712-0033 Koester Anna M. 1
Black Hannah L. 3
Laub Marco 1
Miller Lucy 1
http://orcid.org/0000-0001-8715-398X Heffernan Linda 4
http://orcid.org/0000-0002-7956-7805 Simpson Jeremy C. 4
Mastick Cynthia C. 5
http://orcid.org/0000-0002-2358-1050 Cooper Jon 2
Gadegaard Nikolaj 2
Bryant Nia J. 3 nia.bryant@york.ac.uk
http://orcid.org/0000-0001-6571-2875 Gould Gwyn W. 6 gwyn.gould@strath.ac.uk
1 Institute of Molecular Cell and Systems Biology, University of Glasgow , Glasgow , UK
2 School of Engineering, University of Glasgow , Glasgow , UK
3 Department of Biology, University of York , York , UK
4 School of Biology & Environmental Science, University College Dublin , Dublin , Ireland
5 Molecular Biosciences, University of Nevada - Reno , Reno, NV , USA
6 Strathclyde Institute of Pharmacy and Biomedical Sciences, University of Strathclyde , Glasgow , UK
Moraczewska Joanna
Electronic publication date: 2020 Mar 5
Publication date: 2020
Volume: 8
Electronic Location ID: e8751
Received 2019 Dec 16; Accepted 2020 Feb 14
Copyright: © 2020 Morris et al.
Copyright year: 2020
Copyright holder: Morris et al.
License: This is an open access article distributed under the terms of the Creative Commons Attribution License, which permits unrestricted use, distribution, reproduction and adaptation in any medium and for any purpose provided that it is properly attributed. For attribution, the original author(s), title, publication source (PeerJ) and either DOI or URL of the article must be cited.
License URL: https://creativecommons.org/licenses/by/4.0/

Keywords: Membrane, Transport, Insulin, Diabetes, GLUT4, Endosome

Funding: University of Glasgow Diabetes UK BHF Studentship Number FS/16/55/32731 Diabetes UK 13/0004725 and 17/0005605 Irish Research Council European Research Council through the Consolidator Award “FAKIR” This work was supported by a Lord Kelvin/Adam Smith PhD studentship from the University of Glasgow to Silke Morris, the Arthur and Sadie Pethybridge studentship from Diabetes UK to Jessica Sadler, BHF studentship number FS/16/55/32731 to Anna Koester and grant 13/0004725 and 17/0005605 from Diabetes UK to Nia Bryant and Gwyn Gould. Linda Heffernan was supported by a postgraduate scholarship from the Irish Research Council. Nikolaj Gadegaard acknowledges support from the European Research Council through the Consolidator Award “FAKIR”. The funders had no role in study design, data collection and analysis, decision to publish, or preparation of the manuscript.

==============================
Insulin-stimulated glucose transport is a characteristic property of adipocytes and muscle cells and involves the regulated delivery of glucose transporter (GLUT4)-containing vesicles from intracellular stores to the cell surface. Fusion of these vesicles results in increased numbers of GLUT4 molecules at the cell surface. In an attempt to overcome some of the limitations associated with both primary and cultured adipocytes, we expressed an epitope- and GFP-tagged version of GLUT4 (HA–GLUT4–GFP) in HeLa cells. Here we report the characterisation of this system compared to 3T3-L1 adipocytes. We show that insulin promotes translocation of HA–GLUT4–GFP to the surface of both cell types with similar kinetics using orthologous trafficking machinery. While the magnitude of the insulin-stimulated translocation of GLUT4 is smaller than mouse 3T3-L1 adipocytes, HeLa cells offer a useful, experimentally tractable, human model system. Here, we exemplify their utility through a small-scale siRNA screen to identify GOSR1 and YKT6 as potential novel regulators of GLUT4 trafficking in human cells.

Introduction

Insulin-stimulated glucose transport in peripheral tissues is largely achieved by the delivery of intracellular glucose transport-4 (GLUT4)-containing vesicles to the cell surface where they dock and fuse (Bryant, Govers & James, 2002; Jaldin-Fincati et al., 2017; Klip, McGraw & James, 2019). This increases the numbers of functional transporters at the cell surface thereby raising the Vmax for glucose entry into the cell (Klip, McGraw & James, 2019). Insulin-stimulated glucose transport is impaired in Type-2 diabetes, providing a significant impetus into understanding the molecular mechanisms involved in this event (Kahn, 1992; Garvey et al., 1993, 1998; Klip, McGraw & James, 2019).

Many of the studies which investigate the mechanism of GLUT4 translocation use adipocytes either from animal or human tissues, or the 3T3-L1 adipocyte cell model. Primary tissue suffers from difficulty in employing many routine molecular manipulations (e.g. RNAi, over-expression, etc.) as cells de-differentiate in culture over time, and the 3T3-L1 adipocyte cell line is difficult to transfect, requiring usually either electroporation or viral nucleic acid delivery mechanisms (Orlicky & Schaack, 2001; Puri et al., 2007). Furthermore, 3T3-L1 adipocytes require extensive periods in culture which can make high throughput screening difficult.

Recently, we and others have established HeLa cells expressing epitope-tagged GLUT4 as a model system which facilitate studies into aspects of GLUT4 biology by virtue of the ease of genetic manipulation of this commonly employed cell line (Kawase et al., 2006; Haga, Ishii & Suzuki, 2011; Kioumourtzoglou et al., 2015; Gulbranson et al., 2017; Camus et al., 2020). Here, we describe characterisation of these cells compared to 3T3-L1 adipocytes expressing the same GLUT4 reporter. We show that these two lines, while differing in the magnitude of the insulin response, show many similar characteristics. While we advocate working on cell types as close to physiologically relevant tissues as possible, we nevertheless believe our beta-testing analysis of HeLa cells expressing GLUT4 indicate these cells provide a useful system for initial trials or larger scale screening analyses which cannot be readily undertaken in either primary cells or differentiated cell culture systems. In support of this, we present data analysing the effects of vSNARE knockdown on GLUT4 trafficking and a small-scale RNAi screen for novel effectors of GLUT4 trafficking that suggest an important role for GOSR1 and YKT6 on GLUT4 trafficking and stability. We conclude that HeLa cells are a useful model for preliminary or high-throughput studies of GLUT4 traffic.

Materials and Methods

Cells

HeLa and 3T3-L1 cells were obtained from the ATCC. A HeLa cell line stably expressing HA–GLUT4–GFP was created following infection with a lentiviral construct encoding GFP-tagged GLUT4 carrying an HA epitope in the first extracellular loop; clones were isolated by limited dilution (Muretta, Romenskaia & Mastick, 2008; Muretta & Mastick, 2009). 3T3-L1 adipocytes expressing the same construct were generated from 3T3-L1 fibroblasts engineered to express this construct stably as outlined (Muretta, Romenskaia & Mastick, 2008). Cells were grown and differentiated as previously described (Muretta, Romenskaia & Mastick, 2008; Sadler, Bryant & Gould, 2015). Cells were plated on a Total Internal Reflection Microscopy (TIRFM)-compatible observation chambers (Ibidi, Glasgow, Scotland) which contained a 170 µm coverslip base. HeLa cells were plated to a density of 4 × 103 cells per chamber to distinguish single cells from the population for observation. Cells were left to attach to the surface of the coverslip overnight. Prior to measurement, all cells were serum starved for 2 h.

TIRFM

Total Internal Reflection Microscopy images were acquired using an in-house constructed objective based TIRFM system. The light from a 481 nm diode laser (HORIBA) was directed to the far aperture of a 1.45 NA objective (Zeiss) using a Till Photonics TIRF condenser. The condenser contains a micrometre screw gauge for lateral manipulation of the beam relative to the optical axis. The resultant fluorescence light was collected by the same objective and focussed to an Andor Ixon EMCCD using a C-mount 1.6x expansion lens. The penetration depth of the evanescent field was measured using 10 µm fluorescent particles and was found to be 110 nm. For discrete membrane intensity imaging a series of 10 individual frames were acquired at each time point with an exposure time of 500 ms. For all time-lapse image sets the rate of acquisition was set to two frames per second.

Live cell imaging

All cells were imaged in Ibidi chambers as above. A temperature of 37 °C was maintained using a temperature control microscope insert (PeCon, Erbach, Germany). Insulin stimulation was achieved by replacing 50% of the chamber’s media with media containing 200 nM insulin to provide an overall concentration of 100 nM.

Membrane associated fluorescence image analysis

All image analysis was performed using the ImageJ/Fiji platform. For imaging of membrane associated fluorescence intensity, a perimeter was defined around the footprint of an individual cell for the initial images prior to insulin stimulation. Within this footprint the average pixel-wide fluorescence intensity was measured at discrete time points before and after the addition of 100 nM insulin.

Mobile and static vesicle analysis

Images were processed as follows: The signal from GLUT4 vesicles versus the uneven diffuse fluorescent background was enhanced through the implementation of a rolling ball algorithm (Sternberg, 1983). Subsequently, noise was removed through the use of the ‘de-speckle’ and ‘outlier removal’ subroutines of ImageJ. To differentiate between static and mobile vesicles located at the membrane, image stacks were accumulated for 2 min prior to insulin stimulation and for 25 min after. Stacks for individual cells were segmented to 1 min time bins for analysis. The average projection image was generated for each 1 min stack and was subtracted pixel by pixel from each image of the original stack. The resultant image stack contained data relating only to moving GLUT4-containing vesicles. This secondary image stack was subsequently subtracted from the original stack to provide a series of images containing only stationary vesicles.

Individual vesicles were defined by the following criteria using the FindFoci algorithm (i) that the fluorescent point had a local maxima value, (ii) 75% of the peak intensity was contained within a 5 pixel radius and (iii) the point was larger than a minimum 2 pixel radius (Herbert, Carr & Hoffmann, 2014).

siRNA transfection

The day prior to transfection, cells were plated onto glass coverslips in a 24-well plate at a density of 7,500 cells/well. The following day, cells were transfected with 200 nM SMARTpool siRNAs (GE Healthcare Ltd., Chicago, IL, USA) specific for VAMP isoforms or Syntaxin-16, as indicated in the figure legends using DharmaFECT (ThermoFisher, England, UK), according to the manufacturer’s instructions (Simpson et al., 2007; Simpson, 2009). Cells were assayed between 48 and 72 h after transfection. Prior to use, cells were incubated in serum-free media for 2 h; 1 µM insulin was added for a further 20 min as indicated on the figure legends.

For cell lysates, transfected cells were washed twice with ice-cold PBS then solubilised in RIPA buffer: 20 mM Tris-HCl (pH 7.5), 150 mM NaCl, 1 mM Na2EDTA, 1 mM EGTA, 1% (v/v) NP-40, 1% (v/v) sodium deoxycholate, 2.5 mM sodium pyrophosphate, 1 mM β-glycerophosphate, 1 mM Na3VO4 and proteinase inhibitors. Anti-VAMP2 (#104 202), anti-VAMP3 (#104 103), anti-VAMP4 (#136 002), anti-VAMP5 (#176 003), anti-VAMP7 (#232 003) and anti-VAMP8 (#104 302) were from Synaptic Systems, Germany; gels and blots were performed as in Sadler, Bryant & Gould (2015).

For the screening experiments, the day prior to transfection 9,000 HeLa cells expressing HA–GLUT4–GFP were plated onto each well of a glass bottomed 96-well plate. Cells were transfected with 3 μM siRNA using the transfection reagent Lipofectamine 2000 and Opti-MEM reduced serum medium according to the manufacturer’s instructions. After 4 h exposure to siRNA, growth media was added to each well and cells were incubated for 48 h prior to fixation; cells were washed in PBS containing DAPI to visualise nuclei and representative fields of cells (five from each well, performed from four experimental replicates). Images were collected using a 63x/1.4NA objective and analysed using ImageJ/FUJI software. Details of the siRNA sequences employed are provided in Table S1.

HA–GLUT4–GFP translocation: immunofluorescence

HeLa cells were washed three times in ice-cold PBS then fixed in 3% para-formaldehyde (PFA) for 20 min. After quenching and washing in PBS containing 1% BSA and 5% goat serum, cells were incubated with anti-HA monoclonal antibody (Covance Research Products t# MMS 101P) at 1:500 for 45 min at room temperature, then washed three times and surface bound monoclonal antibody detected using Alex-Fluor labelled secondary antibodies (1:200). Cells were washed and mounted using Immuno-mount and imaged using a Zeiss Pascal unit. Images were collected using a 63x/1.4NA objective and analysed using ImageJ/FUJI software. Typically, between four and six random fields of view were captured from each coverslip, and each experiment replicated four times on independent passages of cells. Note that within a particular experiment, the intensity and pinhole settings on the confocal were kept constant to allow direct comparison between experimental conditions. Fluorescence intensity values from ImageJ were expressed as an HA/GFP ratio and the value in unstimulated cells set = 1.0 to allow comparison between independent experiments.

HA–GLUT4–GFP translocation: FACS

HA–GLUT4–GFP HeLa cells were seeded onto six well plates (~300,000 cells/well) 24 h pre-analysis. On the experiment day cells were serum-starved for 2 h and half of the HA–GLUT4–GFP HeLa cell samples were stimulated with 1 μM insulin for 40 min at 37 °C. The plates were then placed on ice where all subsequent steps were performed with use of ice-cold solutions. Surface GLUT4 was detected by immunostaining with anti-HA antibodies in intact cells. Cells were incubated with labelling medium containing primary anti-HA antibody at a concentration of 1:200 in DMEM with 10% (v/v) FCS for 1 h. Cells were washed 3 times with PBS and incubated in labelling medium containing 1:300 secondary antibody conjugated with AlexaFluor® 647 for 1 h. Cells were washed with PBS and gently dissociated with collagenase type I (2 mg/ml (w/v)) in PBS supplemented with 0.5 mM EDTA and 10% (v/v) FCS at 37 °C for 10 min. Samples were diluted in PBS and gently filtered through a 100 μm cell strainer to remove clumps of cells and analysed on a BD™ LSR II flow cytometer. Events of 50,000 cells were collected for each experimental condition. Identical methodology was employed when assaying translocation in 3T3-L1 adipocytes expressing HA–GLUT4–GFP by FACS. In both cases, cells not expressing HA–GLUT4–GFP or stained with HA-antibodies/secondary antibodies were used to determine background values. Fluorescence intensities were calculated form geometric means.

Co-localisation assays

HA–GLUT4–GFP HeLa cells were cultured on 96 well plates with a glass bottom. Prior to staining, they were washed 3x with PBS and fixed with PFA for 20 min at room temperature. Cells were washed 3x with PBS and incubated in permeabilisation buffer (PBS containing 0.1% w/v Triton X100) for 4 min, then washed 3x with PBS and blocked with PBS containing 0.2% (w/v) fish skin gelatine and 0.1% (v/v) goat serum for 30 min. Cells were stained by incubating with the primary antibody recognising a marker for the ER, the Golgi, or the ERGIC for 60 min. and with a secondary Alexa Fluor 568 tagged antibody for 45 min. Nucleus staining was carried out by incubation with 1 μg/ml DAPI for 5 min. Representative fields of cells (four from duplicate coverslips for each condition, repeated four times) were imaged using a Zeiss Pascal unit. Images were collected using a 63x/1.4NA objective and analysed using ImageJ/Fiji software and the JaCoP plugin (Bolte & Cordelières, 2006).

Statistical analysis

Statistical analysis was conducted using Prism software; tests are described in the figure legends or text.

Results

GLUT4 in HeLa cells exhibits insulin-dependent translocation to the cell surface

The majority of current investigations of GLUT4 recruitment to the plasma membrane suffer from a low throughput due to the lengthy isolation and culturing of adipocytes, which are also difficult to experimentally manipulate. As a result, there is a clear need for a robust insulin-sensitive experimental cell model expressing GLUT4 as a useful ‘test-tube’ for initial experiments. Such a system would provide a platform for high throughput investigations. The HeLa cell line is an immortal cervical cancer cell line, originally isolated in 1951, and is the most widely investigated cell model (Macville et al., 1999). While HeLa cells do not contain any endogenous GLUT4, they do possess insulin sensitivity, and exhibit insulin-stimulated phosphorylation of Akt and AS160, two key signalling intermediates in insulin-stimulated GLUT4 translocation (Camus et al., 2020).

We therefore generated stable clones of HeLa cells expressing HA–GLUT4–GFP. In the absence of insulin this construct was intracellularly sequestered and was present in intracellular peripheral vesicles and within a large perinuclear depot a distribution highly similar to that observed in 3T3-L1 adipocytes (Fig. 1). This well-characterised construct allows for detection of total GLUT4 levels (using the GFP signal) and the quantification of cell surface exposed molecules via the HA epitope inserted into the large exofacial domain between transmembrane helices I and II. This construct has been extensively validated in numerous laboratories (Lampson et al., 2000; Eyster, Duggins & Olson, 2005; Eyster et al., 2006; Muretta, Romenskaia & Mastick, 2008; Zhao et al., 2009; Lizunov et al., 2012). Figure 1 show the effect of insulin on HA–GLUT4–GFP translocation in HeLa cells (Fig. 1A) compared to 3T3-L1 adipocytes (Fig. 1B). In both cell types, a robust insulin-stimulated delivery of GLUT4 to the cell surface is observed. Figures 1A and 1B shows a typical set of confocal images in which the extent of translocation is revealed by the increase in HA staining upon insulin stimulation, and Fig. 1C shows the result of quantification of insulin-stimulated HA–GLUT4–GFP translocation using FACS. The latter allows quantification of many thousands of cells per condition and is the most reliable method for quantifying translocation in either cell type as a result. However, image analysis of fluorescence intensity of confocal images can also be readily used to quantify translocation—see further examples below. Insulin-stimulated translocation of HA–GLUT4–GFP to the cell surface was inhibited robustly by 50 nM wortmannin in both cell types (data not shown) (Clarke et al., 1994; Wang et al., 2019), suggesting similar insulin-dependent signalling processes underly these responses. Others have reported similar magnitudes of translocation using HeLa cells expressing HA–GLUT4–GFP across several studies of this type, insulin elicited a 2.5–3 fold increase in cell surface GLUT4 staining (Haga, Ishii & Suzuki, 2011; Kioumourtzoglou et al., 2015; Gulbranson et al., 2017; Wang et al., 2019). The roughly 3-fold increase in GLUT4 translocation in HeLa cells is broadly comparable with other human tissues—for a recent review of this issue see N.J. Bryant & G.W. Gould, 2020, unpublished data.

Figure 1 HA–GLUT4–GFP translocation in HeLa cells and 3T3-L1 adipocytes.

(A) HeLa cells stably expressing HA–GLUT4–GFP were incubated without (Basal) or with 100 nM insulin for 1 h in serum-free media, washed fixed and stained for cell surface GLUT4 using the exofacial HA-epitope as described in “Materials and Methods”. Shown are representative fields of cells in which the GFP moiety is pseudo-coloured green and the HA staining is pseudo-coloured blue. A merged image is presented as shown. The data shown is typical of more than 10 experiments of this type on different batches of stable HeLa cell clones expressing HA–GLUT4–GFP. (B) 3T3-L1 cells stably expressing HA–GLUT4–GFP were incubated with or without (Basal) 100 nM insulin for 20 min in serum-free media, washed fixed and stained for cell surface GLUT4 using the exofacial HA-epitope as described in “Materials and Methods”. Shown are representative fields of cells exactly as outlined in (A). Data from a representative experiment is shown, replicated on four different batches of stable cell clones. (C) Translocation of HA–GLUT4–GFP to the cell surface in HeLa cells and 3T3-L1 adipocytes was assayed using FACS. Shown is the change in HA/GFP signal in response to insulin, expressed relative to the basal, in n = 3 experiments for each of the cell types shown with 50,000 cells per condition. A significant increase in cell surface GLUT4 levels was detected in both cell types, *p < 0.05 and **p ~ 0.01.

Insulin-stimulated delivery of GLUT4 into the TIRF zone

Time-lapse live cell TIRFM was employed to quantify mobile and stationary vesicles located adjacent to the plasma membrane following insulin stimulation in both cell types.

We first quantified the extent of translocation by measuring the time-dependent increase in GFP signal in the TIRF zone (a typical data set for 3T3-L1 adipocytes is shown in Fig. 2A). Both analyses reveal that insulin stimulates translocation of HA–GLUT4–GFP to the surface, but that HeLa cells exhibit a smaller response than 3T3-L1 adipocytes, 1.89 + 0.4-fold versus 3.3 + 0.85-fold. Note that the magnitude of the insulin response in these experiments is likely underestimated; quantification of the GFP signal does not represent only GLUT4 in the plasma membrane but will also report GLUT4 vesicles in the TIRF zone that are not fused with the plasma membrane. Figure 2B shows that the rate of translocation of GLUT4 in these cells exhibited half-times of 12.3 + 2.2 min in adipocytes (n = 15 cells) and 17.1 + 6.3 min in HeLa cells (n = 12). The value measured in 3T3-L1 adipocytes is somewhat slower than has been reported by others (5–10 min, see Bogan, McKee & Lodish (2001) and Govers, Coster & James (2004)). The slower rate of translocation in observed in our studies in 3T3-L1 adipocytes and HeLa cells may reflect a slower accumulation of total vesicles into the TIRF zone compared to levels of GLUT4 in the plasma membrane (Gibbs, Lienhard & Gould, 1988; Subtil et al., 2000; Coster, Govers & James, 2004; Martin, Lee & McGraw, 2006; Gonzalez & McGraw, 2006; Muretta, Romenskaia & Mastick, 2008; Muretta & Mastick, 2009; Xiong et al., 2010). This may also in part be a reflection of the temperature homeostasis on the stage being less than ideal due to the home-built nature of the incubation system; nevertheless, these data indicate that insulin-stimulated translocation of GLUT4 in these cell types are broadly comparable.

Figure 2 Translocation of HA–GLUT4–GFP assayed by TIRFM.

HA–GLUT4–GFP expressing 3T3-L1 adipocytes were serum-starved for 2 h and mounted on a heated stage in a home-built TIRF system. Images corresponding to GFP fluorescence were collected prior to insulin addition (0 min) then at the indicated times after addition of 100 nM insulin. Scale bar: 20 µm. Data from a representative experiment is shown in (A). (B) Quantification of the time course of insulin-stimulated increase in GFP fluorescence in the TIRF zone in either HeLa or 3T3-L1 adipocytes. (C) The magnitude of the increase in GFP signal in the TIRF zone upon exposure of the cells to 100 nM insulin. For both (B) and (C), the data is the mean + SEM of 12 HeLa cells and 15 3T3-L1 cells imaged at each time point from at least three biological replicates. *Statistically significant compared to basal p = 0.05; **p = 0.01 statistical significance analysed by 2-way ANOVA.

Comparison of GLUT4 vesicle movement near the plasma membrane

Stacks of time-lapse images were separated into 1-min segments to determine the time-dependent nature of vesicle dynamics as described in “Materials and Methods”. GLUT4-containing vesicles were identified and the time dependent vesicle dynamics were compared for both cell lines. These values were measured for three individual cells within ten different 100 µm2 regions of interest across three separate platings of cells (Fig. 3A). In adipocytes, at the point of insulin stimulation, t = 0, a notable increase in mobile GLUT4-containing vesicles was observed (note t = 0 constitutes the first ~60 s after insulin addition). This period of increased mobility lasted ~5 min before returning to a rate similar to that prior to insulin stimulation. This is in contrast to the behaviour of static GLUT4 vesicles. This is consistent with Fujita et al. (2010) who observed an insulin-dependent increase in the number of rapidly moving GLUT4-containing vesicles approaching the plasma membrane in 3T3-L1 adipocytes, and an insulin-dependent increase in their tethering.

Figure 3 Quantification of vesicle dynamics in the TIRF zone.

Counts of mobile and stationary vesicles for 3T3-L1 adipocytes (A) and HeLa cells (B) stimulated by 100 nM insulin were determined as outlined in “Materials and Methods”. A total of 10 individual 100 µm2 regions of interest were analysed from three cells from three separate platings of cells. Error bars correspond to standard deviation for 30 measured ROIs. Images were recorded at a frame rate of 2 Hz for 15 min where time point 0 corresponds to point of insulin addition.

Stenkula et al. (2010) and Lizunov et al. (2013a) report a dramatic increase in the rate of vesicle fusion with the plasma membrane between 1 and 5 min post insulin. The observed increase in dynamic GLUT4-containing vesicles over a similar time frame in this study would suggest vesicle activity in line with the previously presented kinetic model (Stenkula et al., 2010; Lizunov et al., 2013a). The gradual increase in stationary GLUT4-containing vesicles was predicted as vesicles continue to tether and fuse to the membrane upon stimulation (Fujita et al., 2010), and several studies have reported an insulin-stimulated increase in vesicle tethering (Lizunov et al., 2005, 2009, 2013b; Bai et al., 2007; Stenkula et al., 2010).

In HeLa cells, the quantity of mobile GLUT4-containing vesicles underwent a similar insulin-dependent increase (Fig. 3B). The duration of this increased activity was observed to last ~8 min after insulin stimulation. After the increase in activity the number of mobile GLUT4-containing vesicles returned to a density of roughly 2 per 100 µm2. The slightly longer duration and lower final density is consistent with the extended τ1/2 noted above for insulin-stimulated GLUT4 translocation. One potentially interesting point of divergence between the two cell types is that the density of static GLUT4-containing vesicles decreased in response to insulin in HeLa cells (Fig. 3B) whereas this remained static or even increased in 3T3-L1 adipocytes. This may explain the lower fold increase in GLUT4 translocation in HeLa cells. The explanation for this is not clear, but it is worth noting that the larger more rounded phenotype of 3T3-L1 adipocytes mean that the area of the cytoplasm sampled by TIRF is likely a smaller fraction than that sampled in HeLa cells. These observations support the hypothesis that behaviours of GLUT4-containing vesicles differ subtly between different cell types, and that this may explain variations in the magnitude of GLUT4 translocation. Nevertheless, these data demonstrate that insulin-dependent mobilisation of GLUT4 to the cell surface is an inherent, common, property of a diverse array of cell types.

Common trafficking pathways for GLUT4 in HeLa and 3T3-L1 adipocytes

Previous studies have established important roles for VAMP2 and VAMP4 in GLUT4 trafficking in adipocytes (Williams & Pessin, 2008; Zhao et al., 2009; Sadler, Bryant & Gould, 2015). VAMP2 plays an important role in insulin-dependent GLUT4 translocation to the cell surface, and VAMP4 in the delivery of newly synthesised GLUT4 into the intracellular GLUT4-storage vesicle compartment (Williams & Pessin, 2008). To further test the validity of HeLa cells expressing HA–GLUT4–GFP as a model, we recapitulated the above experiments in this system. For this analysis, we quantified effects by measuring the fluorescence intensity of confocal images (see “Materials and Methods”), as FACS analysis was not practical for reasons of scale/cost. Representative confocal images and quantitation of these experiments are shown in Fig. 4 (blots of lysates from knockdown cells are shown in Fig. S1).

Figure 4 The effect of knockdown of VAMP isoforms on HA–GLUT4–GFP translocation.

HeLa cells expressing HA–GLUT4–GFP were grown and transfected with 200 nM scrambled control sequence (SCR), VAMP2, 4 or 8 SMARTpool siRNA as described. Cells were serum-starved (basal) before being treated with or without 1 µM insulin (insulin) for 20 min. Cells were fixed and surface HA was stained as described. (A) Immunofluorescence images of a typical field for each condition are shown. (B) The fold-change in the HA/GFP ratio with insulin-stimulation. Values were compared using Student’s t-test (*p < 0.05 and **p > 0.05) and are means ± SD of 16 random fields of view, taken from four independent experiments. In this analysis, the HA/GFP ratio in the absence of insulin is set equal to 1 for each siRNA. VAMP2 knockdown significantly inhibited insulin-simulated HA–GLUT4–GFP translocation, *p = 0.01. The apparent reduction in translocation upon VAMP4 knockdown did not reach statistical significance. (C) Quantifies the basal HA/GFP ratio for each VAMP knockdown compared to that observed in SCR siRNA treated cells; values represent the means ± SD of 16 fields of view, taken from four independent experiments **p < 0.05.

We quantified the ability of insulin to increase plasma membrane GLUT4 levels after siRNA treatment, Fig. 4B, in which the basal (unstimulated levels) of plasma membrane GLUT4 are set at 1. In these experiments scrambled siRNA treated cells exhibited a robust insulin response (~3-fold increase). Figure 4B shows that knockdown of VAMP2 significantly impaired insulin-stimulated GLUT4 translocation to the cell surface. Consistent with published results (Williams & Pessin, 2008) we observed no impairment of HA–GLUT4–GFP translocation in VAMP8 (Figs. 4A and 4B) or VAMP3, VAMP5 or VAMP7-depleted cells (Fig. S2). However, we observed insulin-independent increases in HA–GLUT4–GFP levels in the plasma membrane upon VAMP2 and VAMP4 knockdown; this was larger following VAMP2 depletion than VAMP4 (2.7-fold versus 1.8-fold; Fig. 4C). VAMP2 knock down completely abolished any increase in GLUT4 in the plasma membrane following insulin stimulation. Conversely, following VAMP4 depletion, insulin still appeared to increase GLUT4 in the plasma membrane, although the increase did not reach statistical significance (Fig. 4C; ~1–7-fold, compared to ~3-fold in control cells, p = 0.18). Direct comparison of our data with that in 3T3-L1 adipocytes is difficult, as different methodologies have highlighted areas of contention. Williams and Pessin observed reduced levels of cell surface GLUT4 in VAMP2-depleted cells, with no increase in basal levels (Williams & Pessin, 2008). Zhao et al. (2009) report no effect of VAMP2 knockout on GLUT4 translocation, but the preponderance of evidence argues that VAMP2 is the key vSNARE for GLUT4 translocation (Bogan, 2012; Sadler, Bryant & Gould, 2015; Klip, McGraw & James, 2019). It is interesting though that in the muscle cell line L6, tetanus toxin-induced VAMP2 cleavage resulted in enhanced basal GLUT4 levels (Randhawa et al., 2000), a result strikingly similar to that reported here.

These data might be taken to imply that in human cells, GLUT4 passes through a VAMP2-dependent trafficking step, whereas not all GLUT4 passes through a VAMP4-dependent trafficking step. This explanation is in line with the data of Williams and Pessin, who found that VAMP4 was required for the sorting of newly synthesised GLUT4 into the GLUT4-storage compartment (Williams & Pessin, 2008). As VAMP2 and VAMP4 act at different trafficking steps, depletion of either protein affects GLUT4 localisation differently. Consistent with this, VAMP2 has been proposed to regulate GLUT4-containing vesicle fusion with the plasma membrane and VAMP4 to regulate sorting into the GLUT4 storage compartment (Randhawa et al., 2000; Williams & Pessin, 2008; Sadler, Bryant & Gould, 2015). However, the potential that (for example) VAMP2 may regulate GLUT4 trafficking in other endosomal compartments, as has been suggested for example for early to recycling endosome cargo traffic (Aikawa et al., 2006; Ma & Burd, 2019), is not unreasonable. The relative balance of different endosomal pathways may differ between these cell types and thus reveal different facets of GLUT4 trafficking. Consistent with this, studies have suggested that GLUT4 trafficking differs between human and murine cells (Camus et al., 2020), and hence some distinctions between these two cell systems are not entirely unsurprising.

These data suggest that HeLa cells can offer clear insight into GLUT4 trafficking pathways in a robust fashion. Using this system, rapid screening of the VAMP isoforms would have quickly focussed upon VAMP2 and VAMP4, isoforms which are known to contribute to GLUT4 trafficking in 3T3-L1 adipocytes (Williams & Pessin, 2008). Such studies argue that HeLa cells can offer a reliable and rapid screening system for analysis of GLUT4 trafficking which can be used to focus on key targets for further studies in less tractable cell systems. We therefore attempted to perform such a screen.

Identification of new components of GLUT4 trafficking

The use of RNAi screens has opened up new avenues of investigation (Simpson et al., 2007; Simpson, 2009). siRNA in adipocytes, although achievable, is difficult, often requiring multiple rounds of transfection and is thus not easily transferred to larger screening platforms. Hence, we have examined the use of HeLa cells as a tool for screening using transfection of a library of siRNAs targeting a sub-set of SNARE proteins.

Recent studies have suggested that movement of GLUT4 from the ERGIC towards the GLUT4-storage compartment plays a key role in the trafficking of newly synthesised GLUT4 in human cells (Camus et al., 2020). The SNARE proteins BET1, BET1L, GOSR1, GOSR2, SEC22A, SEC22B, SEC22C, Stx5 and YKT6 are known to be involved in trafficking to or from the ERGIC (Zhang & Hong, 2001; Appenzeller-Herzog & Hauri, 2006; Adnan et al., 2019; Linders et al., 2019). We therefore examined the effect of knockdown of these SNAREs on the distribution of HA–GLUT4–GFP by examining overlap of GFP with the ERGIC marker ERGIC-53 or the Golgi marker GM130. In addition, the GFP signal was used as an indication of GLUT4 levels. Details of the SMARTpools are provided in Table S1. The results are shown in Figs. 5A–5D. The subtle changes observed in these experiments highlight a major advantage in using HeLa cells. The ability to screen the required large numbers of cells allowed statistical significance to be established even for small changes in signal overlap. The plastic nature of membrane protein trafficking pathways predicates that interference with one pathway may be compensated for by another, resulting in subtle changes in steady-state localisation.

Figure 5 GLUT4 trafficking involves GOSR1 and Ykt6.

HeLa cells expressing HA–GLUT4–GFP were grown and transfected with 200 nM scrambled control sequence (−VE CON) or SMARTpool siRNA targeting the indicated SNARE proteins. A total of 48 h later, cells were fixed and stained with organelle-specific antibodies (shown in (A) is a stain for the Golgi resident GM130 as a representative example). The extent of overlap between the GFP (GLUT4) signal and ERGIC53 or GM130 was quantified from five random fields each containing 8–12 cells from four independent experiments as described in “Materials and Methods”. (B) Quantification of the ERGIC53/GLUT4 overlap; (C) quantification of the GM130/GLUT4 overlap. In (D), an estimate of total GLUT4 levels was determined by quantification of the GFP/DAPI ratio from the same experimental datasets. In each of (B), (C) and (D), data of all four experiments was normalised to the negative control experiment and 2-way ANOVA with 95% confidence intervals was used for statistical analysis. GOSR1: p = 0.0115 (B). Ykt6: p = 0.0036 (C). GOSR1: p < 0.0001 (D).

Knockdown of GOSR1 (GS28) selectively reduced the overlap of GLUT4 with ERGIC53; none of the other SNAREs examined gave any reproducible changes in overlap (Fig. 5B). GOSR1 participates in ER to Golgi traffic (Xu et al., 2002), but has also been implicated in the transport from early/recycling endosomes to the trans Golgi network (TGN) (Xu et al., 2002; Tai et al., 2004), in a SNARE complex containing Sx5 and YKT6 where it mediates a trafficking route which functions in parallel with Sx16/Sx6 to mediate early/recycling endosome-TGN traffic. Strikingly, depletion of YKT6 enhanced the overlap of GLUT4 with GM130 (Fig. 5C). An important role for these SNAREs is also suggested by increased GLUT4 levels upon GOSR1 knockdown (Fig. 5D); it is important to note therefore that the reduction in ERGIC53 overlap may be underestimated as this analysis did not take into account total GLUT4 levels. These data, although preliminary, are interesting as they reveal a potential role of YKT6 and GOSR1 in GLUT4 trafficking.

Previous work from our group has implicated a Sx16/Sx6 enriched subdomain of the TGN in GLUT4 delivery into the GLUT4-storage compartment (Shewan et al., 2003; Proctor et al., 2006). Figure 5 reveals that this siRNA analysis did not identify statistically significant effects of depletion of Sx6 or Sx16 on either distribution of levels of GLUT4. While both the overlap with ERGIC53 and the GLUT4/DAPI ratio were decreased upon Sx6 knockdown, these did not reach significance (p ~ 0.06 in both cases). This may reflect the relative capacity of the GOSR1/YKT6 pathway compared to the Sx6/Sx16 pathway in different cell types (Camus et al., 2020). These caveats notwithstanding, our data point to a role for the SNAREs GOSR1 and YKT6 in GLUT4 trafficking in human cells. Further work will be required to elaborate the specificity of these effects (is it only GLUT4 traffic that is perturbed?), the role of these SNAREs in more ‘physiological’ cells or tissues such as 3T3-L1 adipocytes where larger magnitudes of insulin-responsive trafficking can be studied. Nevertheless, this data clearly exemplifies the utility of HeLa cells in the rapid identification of potential targets for further study.

Discussion

Here we characterise HA–GLUT4–GFP expressed in HeLa cells and show that these cells exhibit insulin-stimulated GLUT4 translocation with characteristics similar to those observed in 3T3-L1 adipocytes. We conclude that despite some differences between the two cells systems, such as that observed in the patterns of mobile and static vesicles in the TIRF zone, HeLa cells are a useful model for preliminary or high-throughput studies of GLUT4 traffic. Using this system, we confirm and extend previous studies revealing important roles for VAMP2 and VAMP4 in endosomal GLUT4 trafficking and identify GOSR1 and YKT6 as molecules that may control GLUT4 trafficking.

Examination of GLUT4 vesicle dynamics in 3T3-L1 adipocytes have suggested the presence of multiple different GLUT4-containing vesicles that are mobilised to the cell surface. Studies suggest that GLUT4 traffic to the plasma membrane involves small, ~60 nm diameter, GLUT4-containing vesicles, and also at later times after insulin challenge, larger structures, thought to be endosomal in origin (~150 nm) (Lizunov et al., 2005, 2013a; Xu et al., 2011; Chen et al., 2012). Whether both structures exist in HeLa cells, and which contribute to (for example) the mobile and static vesicles examined here remains to be ascertained. A distinction in the relative balance between these compartments in 3T3-L1 adipocytes and HeLa cells may account for the subtle distinctions observed in (for example) the behaviour of static vesicles and the effects of VAMP2 knockdown. Such distinctions though, should not detract from the utility of HeLa cells as a valid model of GLUT4 trafficking in human cells.

Supplemental Information

Supplemental Information 1 Immunoblot analysis of HeLa cells after VAMP knockdown.

HA–GLUT4–GFP expressing HeLa cells were transfected with the indicated siRNA SmartPool (SCR = scrambled control) and lysates prepared as described in “Materials and Methods”. Equal amounts of lysate were separate on SDS-PAGE gels and immunoblotted for levels of the proteins indicated on the left of the figure. VAMP2 levels were decreased by 55 + 18% and VAMP4 levels decreased 82 + 9% in n = 4 experiments of this type.

Click here for additional data file.

Supplemental Information 2 Effect of VAMP3, 5 and 7 knockdown on HA–GLUT4–GFP translocation in HeLa cells.

HeLa cells expressing HA–GLUT4–GFP were grown and transfected with 200 nM scrambled control sequence (Scr), VAMP3, 5 or 7 SMARTpool siRNA as described. Cells were serum-starved (basal) before being treated with or without 1 µM insulin (insulin) for 20 min and cell surface HA staining quantified as outlined in “Materials and Methods”. Values shown are means ± SD of 16 random fields of view, taken from four independent experiments. In each case, insulin induced a statistically significant increase in cell surface GLUT4 staining, but Basal (unstimulated) or insulin-stimulated values did not differ significantly from Scr-treated cells for any of the VAMP knockdowns shown.

Click here for additional data file.

Supplemental Information 3 Raw images, all immunoblots from Fig. S1, data and statistical analysis.

Click here for additional data file.

Additional Information and Declarations

Competing Interests

Author Contributions

Data Availability

Gwyn W. Gould is an Academic Editor for PeerJ.

Silke Morris conceived and designed the experiments, performed the experiments, analysed the data, prepared figures and/or tables, authored or reviewed drafts of the paper, and approved the final draft.

Niall D. Geoghegan conceived and designed the experiments, performed the experiments, analysed the data, prepared figures and/or tables, built TRF microscope, and approved the final draft.

Jessica B.A. Sadler conceived and designed the experiments, performed the experiments, analysed the data, prepared figures and/or tables, authored or reviewed drafts of the paper, and approved the final draft.

Anna M. Koester performed the experiments, authored or reviewed drafts of the paper, and approved the final draft.

Hannah L. Black performed the experiments, authored or reviewed drafts of the paper, and approved the final draft.

Marco Laub performed the experiments, prepared figures and/or tables, and approved the final draft.

Lucy Miller performed the experiments, prepared figures and/or tables, and approved the final draft.

Linda Heffernan performed the experiments, authored or reviewed drafts of the paper, and approved the final draft.

Jeremy C. Simpson conceived and designed the experiments, analysed the data, authored or reviewed drafts of the paper, and approved the final draft.

Cynthia C. Mastick conceived and designed the experiments, analysed the data, authored or reviewed drafts of the paper, and approved the final draft.

Jon Cooper analysed the data, authored or reviewed drafts of the paper, co-wrote grant, and approved the final draft.

Nikolaj Gadegaard conceived and designed the experiments, analysed the data, authored or reviewed drafts of the paper, and approved the final draft.

Nia J. Bryant conceived and designed the experiments, analysed the data, authored or reviewed drafts of the paper, and approved the final draft.

Gwyn W. Gould conceived and designed the experiments, analysed the data, prepared figures and/or tables, authored or reviewed drafts of the paper, and approved the final draft.

The following information was supplied regarding data availability:

Raw data is available in the Supplemental Files.

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
