# Peer review of "Characterisation of GLUT4 trafficking in HeLa cells: comparable kinetics and orthologous trafficking mechanisms to 3T3-L1 adipocytes"

_PeerJ, doi:10.7717/peerj.8751_

## Round 0.1 · original submission · Minor Revisions

· Academic Editor

Minor Revisions

Please respond to the reviewer comments. In particular, please make sure that the statistical methods are clearly described.

·

Basic reporting

Basic reporting standards are mostly met; see comments under 3) Validity of the Findings.

Experimental design

Experimental design standards are mostly met; see comments under 3) Validity of the Findings.

Validity of the findings

This manuscript examines GLUT4 trafficking in HeLa cells and compares it to that in 3T3-L1 adipocytes, and then goes on to use the HeLa cells for an siRNA-based analysis and screen. The data to establish the HeLa cell system are similar to that recently published by Camus et al., J Cell Biol. 2020 Jan 6;219(1). pii: e201812135 and so are not particularly novel, but they do further establish and expand upon the utility of the HeLa cell system for studies of GLUT4. The siRNA data are convincing, although one wishes that the work were taken a little further. Because the guidelines for PeerJ do not require assessment of impact or novelty, this should not preclude publication.

Specific comments:

In Fig. 2, the rate of increase in GFP fluorescence in the TIRF zone is rather slow, compared to previous studies reporting that maximal or near-maximal GLUT4 translocation occurs within 5-10 min after insulin addition, and that a steady state is reached by 15-20 min. Detailed kinetic were published in Govers et al., Mol Cell Biol. 2004 Jul;24(14):6456-66 and Bogan et al., Mol Cell Biol. 2001 Jul;21(14):4785-806. There is some discussion of this point, but the wording is unclear. Line 248 reads: “The slower rate of translocation…” without stating what translocation is slower than. Presumably, translocation is slower than that previously reported, but the text needs to state this more explicitly.

The manuscript uses the term “GSVs” on line 126 and subsequently in several other places, but this term is not defined. The “GLUT4 Storage Vesicles” are thought to be cell type specific, and can be detected in fat and muscle cells, and not in other cell types. Even the recent Camus et al JCB paper reports that CHC22 is involved in a “GLUT4 pathway” that is present in HeLa cells, but is careful not to claim that it mediates budding of the GSVs (or GSC) itself. It would be appropriate to replace this term with “GLUT4-containing vesicles” – which is more noncommittal about what these vesicles actually are.

The static vs mobile GLUT4 vesicle analysis is somewhat confusing, based on the description in the Methods section and on the data shown in Fig. 3. It appears that in 3T3-L1 adipocytes, insulin caused no change in the number of static vesicles in the TIRF zone, and caused a transient increase in the number of mobile vesicles observed. By contrast, in HeLa cells, the numbers of both static and mobile vesicles decreased. These observations do not support the idea that HeLa cells and 3T3L1 cells behave similarly.

Of note, the adipocyte-specific GSVs are small (60 nm) and can be difficult to visualize by TIRF microscopy, and GLUT4 traffics to the plasma membrane in both GSVs and, subsequently, in larger (150 nm) endosome-derived vesicles in 3T3-L1 adipocytes (Xu et al., J Cell Biol. 2011 May 16;193(4):643-53 and Chen et al., J Cell Biol. 2012 Aug 20;198(4):545-60; the Chen reference identified the GSVs as Rab10-positive). It may be that most of the vesicles observed in the Lizunov et al J Cell Biol 2005 article (and possibly subsequent papers) are the larger, endosome-derived vesicles, which constitute the ‘second wave’ of GLUT4 translocation and are more readily detected using TIRF microscopy. These considerations further complicate the interpretation of the static vs mobile GLUT4-containing vesicles in the TIRF zone in HeLa cells. It is unclear what sort of vesicles are present in HeLa cells, and to which type of vesicles they are being compared in 3T3-L1 adipocytes.

It is somewhat disconcerting that Figure 5 does not examine GLUT4 translocation to the plasma membrane, despite that this has been the main focus of figures up until this point. Does insulin stimulation affect the colocalization of GLUT4 with ERGIC53 or GSM130 (or alter the GLUT4/DNA ratio) in HeLa cells? If so, are these effects blocked by GOSR1 or Ykt6 siRNA treatment?

Having identified GOSR1 and Ykt6 as proteins that affect in GLUT4 in HeLa cells, it would be logical to then examine these in 3T3-L1 adipocytes. Indeed, a main point of the manuscript is that HeLa cells can be used for preliminary studies, which would then be tested in a more physiologic cell type. Thus, the conclusion that these proteins are normally involved in GLUT4 translocation would still have to be characterized as preliminary.

Other items:

Methods:

Line 149 do the authors mean 3 micromoles siRNA rather than 3 micromolar siRNA?

For the FACS analysis – how were background fluorescence intensities determined? If the flow cytometer measures fluorescences on a logarithmic scale, then were cells not expressing GFP or not stained for cell surface HA used to assess background? Was the geometric mean or median used to quantify fluorescence intensities?

References:

The Garvey et al 1993 reference is repeated twice.

Additional comments

none

Reviewer 2 ·

Basic reporting

The manuscript is well written and clear throughout, with 1 or 2 exceptions.
1. Please check for use of standard abbreviations in the methods - e.g. minutes, min
2. Please check consistency of units in methods - mg/mL, ug.ml-1

Literature references are appropriate.

Manuscript is structured appropriately, and the figures of good standard.

The results are well written, and provide a good overview of the hypotheses for each subsequent experiment.

Experimental design

The Authors describe a series of comparisons of GLUT4 trafficking in 3T3-L1 adipocytes and HeLa cells. The studies reveal similarities between these cells lines in specific aspects of GLUT4 trafficking. Data included in this manuscript are useful to the field in that they provide a rationale and evidence for the utility of HeLa cells for screening for novel regulators of GLUT4 trafficking. Using HeLas cells, the Authors report that GOSR1 and YKT6 as novel regulators of GLUT4 traffic (in HeLa cells).

The manuscript clearly describes the rationale for the study.

Methods - In general very well described.
Please define the permeabilisation buffer.
Also, include the specific test used to define statistical significance - e.g. figure 2. For tests comparing two values, one where there there is no error associated, did the Authors use a one-sample t-test?

Can the Authors clarify the statement "generated stable clones of HeLa cells" (Results and Discussion). Were single clones of HeLa cells expressing HA-G4-GFP isolated, and the experiments performed in a cloncal cell line? This is not clear from the methods.

Validity of the findings

The data presented include controls and replicates as needed.

Comments:

Figure 1C. The descrption of these data does not talk about the 3T3-L1 response, in comparison to the HeLa cell data. This is a key difference between the 2 cell lines and should be highlighted here. Is the difference in resposne due to higher basal in HeLa cells, or a greater response in L1 cells?

Figure 2. The writing around these data is a little unclear:
"The slower rate of translocation ... GLUT4 in the plasma membrane". What is the translocation "slower" than? Is this in comparison to othe reported T1/2?

One additional piece of data that would be very informative would be a comparison between TfR (endosomal recycling) and GLUT4 in the two cell lines. For example - colocalisation between GLUT4 and TfR in unstimulated conditions. This would adress (to an extent) whether HeLa cells have a bone fide specialised GLUT4 compartment or whetehr GLUT4 traffics with TfR in HeLa cells.

While there are some differences in the responses between cell lines, the Authors generally lean towards pointing out similarities in the Abstract. For Fig. 3, while the data show that the number of static/mobile vesicles are similar, there are stark differences in the responses of both the static and mobile GLUT4 vesicles between L1s and HeLas. The Authors acknowledge this by saying that "insulin-dep mobilisation of GLUT4 to the cell surface is an inherant property of .. cell types". It would be useful to include indictaion that there are differences in GLUT4 trafficking between cells in the Abstract/introduction.

Regarding VAMP2 KD data.
- The VAMP2 KD appears to be minor - what is % KD here (Fig. 1 supp)?
- The major effect is increasing cell surface GLUT4 in unstimulated cells. Is this the same or different from what is described in adipocytes? Also, Figure 4A shows that VAMP2 KD chnages the GFP loclaisation - has this been reported before?
- Other papers have described redundancy in the Vamp requirement in adipocytes - this study should be quoted - Variations in the requirement for v-SNAREs in GLUT4 trafficking in adipocytes. Ping Zhao, Lu Yang, Jamie A. Lopez, Junmei Fan, James G. Burchfield, Li Bai, Wanjin Hong, Tao Xu, David E. James. Journal of Cell Science 2009 122: 3472-3480; doi: 10.1242/jcs.047449

GOSR1 and YKT6. Do the Authors suggest this is a specific effect on GLUT4, or an effect on all protein trafficking to/from the ERGIC.

Can the Authors translate these findings to L1s? As suggested, HeLa cells may be used as a screening cell line, but it would be interesting to know if these proteins regulate GLUT4 in adipocytes/muscle cells (cells that express GLUT4). This is not essential.

As a suggestion, would a table of the experiments performed in this study and how the cell types compare be useful?

---

## Round 0.2 · accepted · Accept

· Academic Editor

Accept

Your revised manuscript has been reviewed by two independent experts. Their comments are attached below. We hope you consider us again for future submissions.

·

Basic reporting

No comment

Experimental design

No comment

Validity of the findings

No comment

Additional comments

The authors have addressed my critiques and I think the article is now acceptable for publication.

As an aside, I respectfully disagree with the authors’ response to a comment from to the other reviewer, who suggested examination of the overlap between the GLUT4 reporter and transferrin receptor (TfR) in HeLa cells. The authors stated that McGraw’s group showed a clear distinction between GLUT4 and TfR in CHO cells and that, in their view, repeating the experiment in HeLa cells will likely show that GLUT4 and TfR “will not significantly overlap” and that the data will not add much to the paper. In fact, the Lampson…McGraw MBoC 2001 and Johnson…McGraw JBC 1998 papers reported substantial overlap of GLUT4/IRAP and TfR in CHO cells. Our work (Yu JBC 2007) found that in 3T3-L1 adipocytes, TUG overexpression drew GLUT4 out of a TfR-positive compartment, whereas TUG disruption caused both 1) ablation of this GLUT4-positive, TfR-negative compartment and 2) redistribution of GLUT4 (but not TfR) to the cell surface. Thus, although CHO (and, likely, HeLa) cells may have a “specialized GLUT4 compartment” to some degree, this compartment is not well developed, compared to that which is present in adipocytes. In 3T3-L1 cells, the compartment becomes more prominent at day 2-4 of differentiation, such that GLUT4 targeting to TfR-negative membranes is increased, basal GLUT4 sequestration is increased (e.g. Bogan MCB 2001 and other articles), and the fold-translocation response is greater. This results at least in part from upregulation of sortilin (as shown by Kandror’s group) and possibly other proteins (e.g. Usp25m, KIF5B). Sortilin is expressed to some extent in HeLa cells, yet it is likely that there would still be substantial overlap of GLUT4 and TfR, even using current microscopy techniques. This is contrary to the authors’ suggestion. Whether the GLUT4-positive, TfR-negative staining in HeLa cells accounts for 5% or 25% or 50% of the overall GLUT4 staining may be useful to know, and could be assessed in future work. The information could help to improve screens using this system. Given the difference in the magnitude of the translocation responses, it is likely that the GLUT4-positive, TfR-negative compartment accounts for a much lower fraction of total GLUT4 in HeLa cells, compared to 3T3-L1 adipocytes (in which it is ~50% of total GLUT4).

The above considerations do not detract from the main point of the authors’ manuscript, which is that HeLa cells can be useful for preliminary or high-throughput studies of GLUT4 traffic. However, it is important to avoid pushing the issue too far. The abstract emphasizes similarities between HeLa cells and adipocytes without noting the difference in the magnitude of translocation response. The authors may wish to note this to provide a more balanced view, and I leave this to their discretion.

Reviewer 2 ·

Basic reporting

Morris et al. characterise GLUT4 trafficking in HeLa cells. They report similarities to GLUT4 trafficking in adipocytes and highlight the HeLa cell model as a useful tool in studying GLUT4.

The Authors have addressed all comments.

Experimental design

The Authors have addressed all comments.

Validity of the findings

The Authors have addressed all comments.